# Non-Invasive Ventilatory Strategies to Decrease Bronchopulmonary Dysplasia—Where Are We in 2021?

**DOI:** 10.3390/children8020132

**Published:** 2021-02-11

**Authors:** Vikramaditya Dumpa, Vineet Bhandari

**Affiliations:** 1Division of Neonatology, Department of Pediatrics, NYU Long Island School of Medicine, NYU Langone Hospital Long Island, Mineola, NY 11501, USA; vdumpa@gmail.com; 2Division of Neonatology, Department of Pediatrics, Cooper Medical School of Rowan University, The Children’s Regional Hospital at Cooper, Camden, NJ 08103, USA

**Keywords:** NCPAP, NIPPV, high flow nasal cannula, chronic lung disease, infant, newborn

## Abstract

Recent advances in neonatology have led to the increased survival of extremely low-birth weight infants. However, the incidence of bronchopulmonary dysplasia (BPD) has not improved proportionally, partly due to increased survival of extremely premature infants born at the late-canalicular stage of lung development. Due to minimal surfactant production at this stage, these infants are at risk for severe respiratory distress syndrome, needing prolonged ventilation. While the etiology of BPD is multifactorial with antenatal, postnatal, and genetic factors playing a role, ventilator-induced lung injury is a major, potentially modifiable, risk factor implicated in its causation. Infants with BPD are at a higher risk of developing complications including sepsis, pulmonary arterial hypertension, respiratory failure, and death. Long-term problems include increased risk of hospital readmissions, respiratory infections, and asthma-like symptoms during infancy and childhood. Survivors who have BPD are also at increased risk of poor neurodevelopmental outcomes. While the ultimate solution for avoiding BPD lies in the prevention of preterm births, strategies to decrease its incidence are the need of the hour. It is time to focus on gentler modes of ventilation and the use of less invasive surfactant administration techniques to mitigate lung injury, thereby potentially decreasing the burden of BPD. In this article, we discuss the use of non-invasive ventilation in premature infants, with an emphasis on studies showing an effect on BPD with different modes of non-invasive ventilation. Practical considerations in the use of nasal intermittent positive pressure ventilation are also discussed, considering the significant heterogeneity in clinical practices and management strategies in its use.

## 1. Introduction

Bronchopulmonary dysplasia (BPD) was first described by Northway et al. in 1967 to describe the inflammatory and fibrotic changes noticed in the lungs of premature infants exposed to mechanical ventilation and hyperoxia [1]. Since that first description, multiple factors have altered the clinical care of preterm infants including the enhanced use of antenatal corticosteroids, surfactant replacement therapy, gentler modes of ventilation, improved nutritional care, and increased survival of extremely premature infants, which led to the characterization of a new type of BPD, whose hallmark is alveolar simplification and impairment of vascular growth [2]. The incidence of BPD is up to 40% among infants born before 28 weeks’ gestation, and up to 80% among those born before 24 weeks’ gestation, with substantial variability among centers and geographic regions [3,4,5]. Studies indicate that the incidence of BPD has remained constant over the last few decades [6,7]. However, a recent study analyzing a large database of premature infants showed that the incidence of BPD has decreased over the last few years, with an increase in the use of non-invasive ventilation during the same period [8]. Mechanical ventilation with subsequent ventilator-induced lung injury (VILI) contributes to BPD. The mechanisms of VILI include damage caused: by excessive stretch of the lung tissue from large gas volumes (volutrauma); by high airway pressures (barotrauma); by repeated alveolar collapse and re-expansion (atelectotrauma); and by release of inflammatory mediators from the injured alveolar epithelium (biotrauma) [9]. Injury to the immature lung can disturb the normal postnatal lung development, potentially leading to BPD, and can have untoward effects on other organs, such as the brain, resulting in poor neurodevelopmental outcomes [10]. In the past, routine intubation of extremely preterm infants was the standard of care. It has now been replaced by early use of nasal continuous positive airway pressure (NCPAP), starting in the delivery room, and selective surfactant use, after large trials showed a benefit with such an approach [11,12]. Early use of non-invasive ventilation minimizes the risk for lung injury and decreases the risk of BPD compared to invasive mechanical ventilation [13,14]. Once limited to the use of NCPAP, clinicians now have access to a wide array of multiple non-invasive ventilator strategies [15]. The current review aims to summarize the available evidence on the use of different non-invasive ventilation techniques, specifically to decrease BPD. The quest for an optimal definition of BPD is ongoing [16], but for the purpose of this review, we have utilized the National Institutes of Health consensus definition from 2001 [17].

## 2. Nasal Continuous Positive Airway Pressure

Early use of NCPAP in the delivery room reduces alveolar collapse and helps to establish functional residual capacity. Multiple large randomized clinical trials (RCTs) comparing NCPAP vs. the routine intubation of preterm infants showed favorable outcomes with reduced lung injury in the NCPAP group [11,12,18,19]. Subsequently, three separate meta-analyses and systematic reviews analyzing the studies comparing NCPAP vs. intubation in preterm infants showed a consistent reduction in the outcome of BPD in the NCPAP groups [13,20,21]. What once started as an acceptable alternative to routine endotracheal intubation, NCPAP has now become the standard of care in the management of spontaneously breathing preterm infants who are at risk for respiratory distress syndrome (RDS). In fact, the practice guideline from the American Academy of Pediatrics states that the early use of NCPAP with subsequent selective surfactant administration in extremely preterm infants results in lower rates of BPD/death when compared with treatment with prophylactic or early surfactant therapy [22]. Despite the overall decreased risk of BPD with its early use, NCPAP still has a high failure rate, with 30–80% of infants in the above RCTs needing endotracheal intubation subsequently [13]. Studies done to predict the variables associated with NCPAP failure revealed, expectedly, that smaller gestational age, lower birth weight, higher fraction of inspired oxygen (FiO_2_) requirement, and abnormal initial blood gas values were associated with higher rates of NCPAP failure [23,24,25]. Identifying this subset early on is critical, as early surfactant use in these infants is beneficial in reducing BPD, compared to its delayed administration [26]. The recommended NCPAP levels typically range from 4–8 cm H_2_O. In a RCT comparing two levels of NCPAP as post-extubation support in preterm infants, the group with higher pressures (7–9 cm H_2_O) had lower extubation failure compared with lower pressures (4–6 cm H_2_O), with no difference in BPD between the groups [27]. A multicenter RCT is ongoing to evaluate the optimum positive end expiratory pressure (PEEP) needed to prevent NCPAP failure in preterm infants [28]. Nasal intermittent positive airway pressure (NIPPV) is the preferred mode of non-invasive respiratory support in extremely preterm infants, after a brief use of NCPAP during the initial period of stabilization in the delivery room. This is discussed in more detail later in the article.

NCPAP systems are classified into constant flow and variable flow systems depending on the technique used to control the gas flow to the patient. Constant flow systems include bubble CPAP, and ventilator CPAP. Variable flow systems utilize flow opposition with fluidic flow reversal during expiration. Despite multiple RCTs [29,30,31,32,33,34,35,36], there is a lack of good evidence to suggest that any one NCPAP system is superior to another in reducing BPD. A recent systematic review and meta-analysis of studies comparing bubble CPAP vs. other forms of CPAP showed less CPAP failure with the use of bubble CPAP compared to ventilator CPAP, and no difference between bubble CPAP and infant flow CPAP. However, there was no significant difference in the outcome of BPD (8 studies, 816 participants, relative risk or RR 0.8, 95% confidence interval or CI 0.53 to 1.21) [37]. Some centers in the US with seasoned experience using bubble NCPAP have demonstrated success, with low NCPAP failure and low BPD rates [38]. The importance of the proper training of the staff involved and diligent nursing care in achieving successful outcomes cannot be overstated. Meticulous attention has to be given in providing consistent and optimal NCPAP delivery to prevent its failure. Finally, the availability of a particular system and the clinician’s familiarity with its use are more important factors than the benefits offered by any particular NCPAP system or device. Bubble NCPAP has the advantage of cost effectiveness and is an ideal NCPAP system for use in developing countries.

## 3. Nasal Intermittent Positive Pressure Ventilation

NIPPV is a type of ventilator-delivered pressure controlled, time-cycled mode, in which peak inspiratory pressures are delivered at a set rate on top of positive end expiratory pressure (PEEP). It mimics endotracheal ventilation, except for the fact that the pressures are delivered through nares. NIPPV works by recruitment of the collapsed alveoli and stabilizing functional residual capacity, resulting in improved gas exchange, increase in tidal and minute volumes, and decrease in work of breathing [39,40,41]. Other reported physiological effects include improved stability of the chest wall, improved pulmonary mechanics, decreased flow resistance, and reduction in thoracoabdominal asynchrony [42,43]. There is better alveolar recruitment with NIPPV, due to the use of a higher mean airway pressure (MAP). NIPPV has also been shown to improve carbon dioxide clearance in lung models and clinical studies [44,45]. Synchronized NIPPV (SNIPPV) refers to the use of NIPPV with synchronization to the patient’s inspiratory efforts. This can be achieved with use of ventilators incorporating flow synchronization mechanisms (currently available in Europe) [46]. In the United States, ventilators able to deliver SNIPPV using a pneumatic capsule placed on the abdomen (Graseby’s capsule) are no longer in production. Non-invasive neurally adjusted ventilator assist (NIV-NAVA) is a different mode of non-invasive ventilation that can deliver SNIPPV, and is discussed separately. SNIPPV leads to better patient–ventilator synchrony, supporting the concept that spontaneous breaths lead to better outcomes [47,48]. Clinical trials have demonstrated the effectiveness of SNIPPV over NIPPV and NCPAP in reducing the need for intubation in RDS, improving the success of extubation, and treating apnea of prematurity, with a reassuring absence of relevant side effects, although no difference in the outcome of BPD was observed [40,45,49,50,51,52]. A detailed review on the synchronization of NIPPV and its benefits can be found in an article by Moretti et al. [53].

The largest RCT done to date, comparing NIPPV to NCPAP as the first-used non-invasive respiratory support in infants <1000 g birth weight (BW) and a gestational age (GA) of <30 weeks, did not show a difference in BPD between the groups (33.9% vs. 31%, odds ratio or OR 1.14 95% CI 0.84–1.54, *p* = 0.32) [54]. However, this was a pragmatic trial with the NIPPV group including infants on bi-level positive airway pressure (BiPAP)/synchronized BiPAP or SiPAP, leading to significant heterogeneity in the devices and degree of support in the NIPPV group. A Cochrane review with meta-analysis of 10 RCTS (*n* = 1061) comparing NIPPV vs. NCPAP used as a primary mode in preterm infants showed a benefit with the use of NIPPV, with less respiratory failure and need for intubation. The outcome of BPD was not significantly different between the two groups (typical RR 0.78, 95% CI 0.58 to 1.06) [55]. Another Cochrane meta-analysis (10 RCTs, *n* = 1431) comparing NIPPV vs. NCPAP used post-extubation, demonstrated a significant reduction in extubation failure and need for re-intubation with the use of NIPPV. Again, there was no significant reduction in rate of BPD with the use of NIPPV (typical RR 0.94, 95% CI 0.80 to 1.10; typical RD −0.02, 95% CI −0.08 to 0.03) [56]. 

However, a recent large network meta-analysis comparing different modes of non-invasive ventilation (35 studies and 4078 infants) used as primary respiratory support in preterm infants with RDS excluded the studies where the modalities of NIPPV and BiPAP/SiPAP were evaluated as similar interventions [57]. In this study, it was noted that NIPPV resulted in lesser incidence of BPD or death when compared to other modes of frequently used non-invasive ventilation. The same authors performed another network meta-analysis (33 studies and 4080 infants) comparing different non-invasive modes used post-extubation, and concluded that the use of SNIPPV is associated with decreased BPD compared to other modes [58]. While there are no large scale RCTs showing a definite benefit with use of NIPPV to decrease BPD, there is enough evidence to recommend its use both as a primary mode, and after extubation, in preterm infants. This superiority of NIPPV over other modes did not consistently translate to a reduction in BPD in clinical studies possibly because of significant variation in the management of using NIPPV. 

The following section offers practical guidance on the use of NIPPV.

1Initiation of NIPPV: Due to the clear advantages of reducing the risk for intubation and extubation failure, we recommended considering NIPPV in the acute phase of RDS. Synchronized NIPPV, if available, is preferred to non-synchronized NIPPV for the reasons mentioned earlier. In moderate to severe RDS, the success of non-invasive ventilation is augmented with early use of surfactant and caffeine. It is beneficial to administer surfactant early (before 2 hours of life) when the infant requires escalation of the support, rather than increasing the support on NIPPV [26]. Table 1 shows the recommended settings for NIPPV based on clinical evidence-based guidelines [59]. The required support varies depending on the type of nasal interface used. Higher settings might be necessary, as recommended by other investigators [15]. Non-invasive ventilation can be administered through a variety of nasal interfaces, short, bi-nasal prongs are recommended [60], but there have been no trials demonstrating the superiority of one interface over the other in decreasing BPD [61].

The primary mode of NIPPV refers to its use soon after birth, with or without a short period (≤2 h) of intubation for surfactant delivery, followed by extubation. The secondary mode refers to its use after a longer period (>2 h to days to weeks) of intubation.

Adapted with permission from: Dumpa V, Bhandari V. Nasal Intermittent Positive Pressure Ventilation. In: Rajiv PK, Vidyasagar D, Lakshminrusimha S, eds. Essentials of Neonatal Ventilation 1st Edition. Copyright Elsevier India 2018.

2Maintenance on NIPPV: Table 2 serves as a guide to adjust the support based on the blood gases. Based on the clinical presentation of the neonate and the blood gas values, settings are to be adjusted to achieve adequate oxygenation and ventilation. In general, peak inspiratory pressure (PIP) is adjusted to achieve adequate chest expansion, and equal and good breath sounds. PEEP is adjusted based on FiO_2_ requirement and work of breathing. It is important to realize that due to increased resistance and the invariable presence of air leakage, the set pressures are not fully delivered to the infant’s lungs during non-invasive ventilation. Hence, it is expected to have a higher PIP and PEEP requirement than in conventional mechanical ventilation to maintain adequate ventilation and oxygenation. NIPPV has the advantage of increasing the mean airway pressure by increasing the PIP (intermittently) and rate rather than just increasing the PEEP, thus theoretically reducing the risk of air leaks, noted with use of higher PEEP when continuously using NCPAP [12,62]. In the trial by Morley et al., even though the median pressure at admission in both control and intervention groups was 8 cm H_2_O, the pressure at which the infants developed pneumothorax was not recorded. We recommend paying close attention to decreasing air leakage through the mouth by applying a chin strap/pacifier before increasing the ventilator support. Other practical suggestions for achieving success with the use of NIPPV are listed in the appendix of the review article by Bhandari [59].

Adapted with permission from: Dumpa V, Bhandari V. Nasal Intermittent Positive Pressure Ventilation. In: Rajiv PK, Vidyasagar D, Lakshminrusimha S, eds. Essentials of Neonatal Ventilation 1st Edition. Copyright Elsevier India 2018.

3Weaning off NIPPV: Consider weaning off from NIPPV to NCPAP when: a) Respiratory distress is improved; b) Episodes of apnea, bradycardia, and desaturations are less frequent; and c) Blood gases are stable, with pH > 7.25, PCO_2_ < 60 with the FiO_2_ being < 0.3. 

Use of NIPPV is not associated with any increased risk of complications compared to NCPAP. Although an increased incidence of gastrointestinal perforations was noted with use of NIPPV in the 1980s [63], a recent Cochrane review comparing NIPPV to NCPAP in preterm infants did not find any difference in gastrointestinal complications between the two groups [64]. In a RCT (*n* = 151) comparing the complications of NIPPV and NCPAP in preterm infants with RDS, there were no gastrointestinal perforations noted in either group [65]. The incidence of nasal injury was higher with use of NCPAP in the Cochrane review by Lemyre et al. (RR 0.11 (95% CI 0.03, 0.41)), but the evidence for this outcome was graded as low quality by the authors [55].

## 4. Bi-Level Cpap

Bi-level CPAP or biphasic CPAP or SiPAP is not to be confused with NIPPV. In BiPAP, variable flow is delivered to provide alternating high and low PEEP levels. The inspiratory times are much longer, with lower respiratory rates, to allow spontaneous breathing. The delta pressure in these modes is low (often around 3–6 cm H_2_O), thus making it a poor choice to treat hypercarbia, compared to NIPPV. A subgroup analysis of a large multicenter RCT comparing outcomes of infants on NIPPV vs. bi-level CPAP did not show a significant difference in the composite outcome of BPD or BPD/death, but morbidity was higher in the bi-level CPAP group [66]. Another RCT comparing BiPAP to NCPAP, used post-extubation, did not show any significant differences in extubation failure or oxygen requirement, at 28 days and at 36 weeks between the groups [67]. At this time, there is not sufficient evidence to recommend its routine use over NIPPV or NCPAP in preterm infants to decrease BPD.

## 5. High Flow Nasal Cannula

In high flow nasal cannula (HFNC), heated and humidified air is delivered through specialized nasal prongs at flow rates generally ranging between 2 and 8 L/min. The mechanism of action of HFNC includes its ability to generate continuous distending pressure, helping to maintain functional residual capacity, provision of gas flow reducing the inspiratory resistance and work of breathing, and washout of nasopharyngeal dead space improving airway conductance.

The actual distending pressure achieved by HFNC is dependent upon factors such as the flow rate, size of the prongs, and size of the infant, and is very unpredictable and difficult to measure. Nevertheless, the advantages of HFNC include the simple interface, which makes it easy to maintain in place, and having less side effects, such as nasal trauma; which have sparked interest in using it after extubation in preterm infants. 

The HIPSTER trial was an international, multicenter RCT (*n* = 564) of preterm infants ≥28 ^0/7^ weeks GA who were randomized to either HFNC or NCPAP as a primary mode of support. The trial had to be terminated early because of a significant difference in the outcome of treatment failure in the HFNC group (25.5 vs. 13.3%). The outcome of BPD was not studied in this RCT [68]. Similarly, another trial comparing the primary mode of HFNC to NCPAP in preterm infants (GA ≥ 28 weeks and BW ≥ 1000 g) with respiratory distress had to be stopped after an interim analysis showed significantly higher treatment failure in the HFNC group (26.3% vs. 7.9%) [69]. In another RCT by Demirel et al., HFNC and NCPAP were shown to have no significant differences in the outcomes of treatment failure or BPD when used as the primary support in infants ≤32 weeks GA (*n* = 107) [70]. Taha et al. did a retrospective analysis of a large database of preterm infants <1000 gm BW (466 neonatal intensive care units or NICUs, *n* = 2487) who received HFNC or NCPAP, and showed that the use of HFNC compared to NCPAP was associated with higher incidence of BPD (56.8% vs. 50.4%, *p* < 0.05). Other respiratory outcomes, such as days to room air and length of hospitalization, were also higher in the HFNC group [71]. In a meta-analysis of four RCTs (*n* = 645) comparing use of HFNC vs. NCPAP after extubation in infants <32 weeks GA, there was no significant difference in the outcome of treatment failure or the secondary outcome of BPD (OR 0.81, 95% CI 0.57–1.16, I^2^ = 0%) between the groups [72]. More recently, a multicenter RCT comparing the outcomes of preterm infants <34 weeks GA (*n* = 372) who were randomized to HFNC vs. NCPAP/NIPPV after extubation revealed a significantly higher treatment failure in the HFNC group (31% vs. 16%). However, the rate of BPD was not significantly different in the two groups (34% vs. 38%, *p* = 0.34) [73]. In contrast, two recent RCTs comparing HFNC to NCPAP as post-extubation support in premature infants showed that HFNC was as effective as NCPAP in preventing extubation failure, without any differential impact on BPD [74,75]. It is important to acknowledge the slight variation in the flow rates used in all the above studies, but the trials examining HFNC as primary mode used 5–8 L/min, and the ones that evaluated its role as post-extubation support used 3–7 L/min. 

Finally, based on the available data, there is not enough evidence to suggest the use of HFNC as either primary respiratory or post-extubation support to decrease BPD in extremely preterm infants <28 weeks GA. Use of HFNC may be considered after extubation in larger GA infants due to its ease of use and less nasal trauma, but its effect on decreasing BPD remains to be proven. 

## 6. Non-Invasive High Frequency Ventilation 

This is a relatively new modality of respiratory support, wherein high frequency breaths are super-imposed on constant positive airway pressure through a nasal interface. The presumed physiological basis for its mechanism is similar to invasive high frequency ventilation, in which low tidal volumes delivered at extremely rapid rates provide constant lung expansion and effectively improve ventilation. A meta-analysis of eight RCTs (*n* = 463) comparing non-invasive high frequency oscillatory ventilation (NHFOV) vs. Bi-CPAP/NCPAP in preterm infants showed that NHFOV significantly removed CO_2_ and reduced the risk of intubation (RR 0.50, 95% CI 0.36–0.70) compared with Bi-CPAP/NCPAP. The authors did not analyze for the outcome of BPD in this study [76]. A RCT by Chen et al. comparing outcomes of infants extubated to NHFOV vs. NCPAP had similar conclusions of decreased risk of reintubation with NHFOV but no difference in the outcome of BPD [77]. Other small RCTs have shown a benefit of improving certain short-term respiratory outcomes with the use of NHFOV, but the outcome of BPD was not studied in these trials due to the relatively larger GA of the study population [78,79]. Clinical trials are underway studying the effects of NHFV on BPD in preterm infants (NCT03558737, NCT03099694), and further evidence is needed before it can be recommended in this population to decrease BPD.

## 7. Non-Invasive Neurally Adjusted Ventilatory Assist

NAVA is another relatively new modality of respiratory support, which can be provided invasively or non-invasively (NIV-NAVA). In this mode, the system detects the electrical activity of the diaphragm using a special feeding tube embedded with electrodes and delivers synchronized, pressure-controlled breaths with the pressure support proportional to the patient’s inspiratory efforts. This ability to synchronize with the patients’ inspiratory efforts allows for decreased work of breathing, improved gas exchange, and earlier extubation [80,81]. Yagui et al. conducted a RCT (*n* = 123) comparing NIV-NAVA vs. NCPAP as the primary support in infants with BW <1500 g. There was no difference noted in the primary outcome of need for intubation prior to 72 hours of life, or for the secondary outcome of BPD, between the two groups [82]. Other small RCTs have shown that NIV-NAVA is as effective as NCPAP in preventing extubation failure, but large RCTs studying the outcome of BPD in extremely preterm infants are needed before NIV-NAVA can be routinely recommended [83,84,85,86,87].

## 8. Conclusions

BPD is a disease of prematurity with multiple factors involved in its etiopathogenesis. Postnatal strategies to decrease BPD must begin in the delivery room. Use of NCPAP for the delivery room stabilization of the preterm infant, selective surfactant replacement therapy (potentially by less invasive surfactant administration techniques), initiation of caffeine therapy soon after birth, and early use of NIPPV in the neonatal intensive care unit are recommended cornerstones in the management of RDS, which are all proven to decrease BPD. We recommend weaning from NIPPV to NCPAP after the acute phase of RDS to minimize lung injury, and to continue NCPAP support until 32–33 weeks post menstrual age to promote lung growth. Further evidence is needed to support the use of HFNC, NHFV, and NIV-NAVA to decrease BPD in extremely preterm infants.

## Figures and Tables

**Table 1 children-08-00132-t001:** Recommended initial and maximum support in primary and secondary modes with extubation criteria prior to secondary mode of nasal intermittent positive airway pressure (NIPPV).

	Primary Mode-Initial Settings	Primary Mode-Maximum Support	Extubation Criteria Prior to Secondary Mode	Secondary Mode-Initial Settings	Secondary Mode-Maximum Support
**PIP (cm H_2_O)**	4 cm above that required for manual ventilation		≤16	2–4 cm above that required on conventional ventilator	
**PEEP (cm H_2_O)**	4–6	6–8	≤5	≤5	6–8
**MAP (cm H_2_O)**		<1000 g–≤14 ≥1000 g–≤16			<1000 g–≤14 ≥1000 g–≤16
**Frequency (per min)**	30	45	15–25	15–25	40
**iTime (s)**	0.4	0.55	0.45	0.45	0.55
**Flow (L/min)**	8–10	12	8–10	8–10	12

PIP: peak inspiratory pressure; PEEP: positive end-expiratory pressure; MAP: mean airway pressure; Ti: inspiratory time.

**Table 2 children-08-00132-t002:** Suggested changes in NIPPV settings based on blood gases/apnea.

	Hypoxemia	Hypercarbia	Hypercarbia + Hypoxemia	Apnea
**Ti * (seconds)**	↑ (max 0.55)	↓ (min 0.4)	↑ (max of 0.55)	0.4–0.5
**Rate * (/minute)**	↑ (max 40)	↑ (max 40)	↑ (max 40)	↑ (max of 40)
**PIP * (cmH_2_O)**	↑	↑	↑	Usually 15–20
**PEEP * (cmH_2_O)**	↑ (max 8)	-/↓	↑ (max 8)	Usually 4–6

Ti: inspiratory time; ↑: Increase; ↓: Decrease; Max: maximum; Min: minimum; PIP: peak inspiratory pressure; PEEP: positive end expiratory pressure; MAP: mean airway pressure. * Variables determining MAP. Adjust these parameters to increase MAP to a maximum of 14 cm H_2_O in <1000 g and 16 cm H_2_O in ≥1000 g weight infants.

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
