# Peer review of "Non-Invasive Ventilatory Strategies to Decrease Bronchopulmonary Dysplasia—Where Are We in 2021?"

_children, 2021, doi:10.3390/children8020132_

Round 1

Reviewer 1 Report

The authors provide a comprehensive and state of the art review on non-invasive ventilation strategies in the preterm infant and how they can reduce BPD disease burden. The review is of high importance and relevant to the topic. The points raised below might help to further improve the ms.

Specific points

The authors provide decisive data on the benefits of NIPPV versus CPAP. It would be worth to mention that there are no more or other side effects associated with this approach.

The authors provide recommendations for NIPPV settings in Table 1 and 2. It would be worth to document the suggested settings by scientific data or name it a personal recommendation. I.e. there are wide differences between the used settings in the US and Europe concerning maximum PIP and inspiratory time. Long inspiratory times might negatively impact on spontaneous breathing and gas exhalation. Low frequencies before extubation might facilitate atelectasis especially when the diameter of the endotracheal tube is small.

The documented action of NIPPV is the increase in mean airway pressure, are there data available that NIPPV leads to improved C02 clearance?

The authors provide detailed information on highflow trials. It would be worth to include differences in highflow settings between the trials in the judgement.

It would be helpful, if the authors could specify their sentence on page 4: “It is beneficial to administer surfactant early (before 2 hours of life) when infant requires escalation of the support.” and add a reference.

Page 2: The sentence “Routine intubation of extremely preterm infants has given the way for early use of nasal continuous positive airway pressure (NCPAP) starting in the delivery room and selective surfactant use, after large trials showed a benefit with such an approach [11,12].” does not come clear-please rephrase.

The authors should avoid to name specific respirators and highflow devices but should instead specify the mode of action of the different devices to stay product independent.

Author Response

The authors provide a comprehensive and state of the art review on non-invasive ventilation strategies in the preterm infant and how they can reduce BPD disease burden. The review is of high importance and relevant to the topic. The points raised below might help to further improve the ms.

Specific points

C1: The authors provide decisive data on the benefits of NIPPV versus CPAP. It would be worth to mention that there are no more or other side effects associated with this approach.

R1: Thank you for the suggestion. We have now included a separate paragraph on page 5 commenting on the complications associated with use of NIPPV and NCPAP.  

C2: The authors provide recommendations for NIPPV settings in Table 1 and 2. It would be worth to document the suggested settings by scientific data or name it a personal recommendation. I.e. there are wide differences between the used settings in the US and Europe concerning maximum PIP and inspiratory time. Long inspiratory times might negatively impact on spontaneous breathing and gas exhalation. Low frequencies before extubation might facilitate atelectasis especially when the diameter of the endotracheal tube is small.

R2: Thank you for the suggestion. We have now provided clarification for the basis of the recommended settings and the reasons why there might be differences among centers. (Page 4).

C3: The documented action of NIPPV is the increase in mean airway pressure, are there data available that NIPPV leads to improved C02 clearance?

R3: Thank you for the interesting question. In the Cochrane reviews, NIPPV was shown to improve respiratory failure compared to NCPAP. Respiratory acidosis was one of the components of respiratory failure in the studies. Therefore, it suggests that NIPPV leads to better CO2 clearance compared to NCPAP. We have also now included references for studies showing improved CO2 clearance with the use of NIPPV. (Page 3 under NIPPV).

C4: The authors provide detailed information on highflow trials. It would be worth to include differences in highflow settings between the trials in the judgement.

R4: Thanks for the valuable suggestion. We have noted that all the studies had fairly similar protocols on escalation and weaning of the flow rate depending upon the clinical status of the patients. Generally, the studies examining HFNC as primary mode used flow rates of 5-8 L/m and the ones evaluating its role as post-extubation support used 3-7 L/min. We included this sentence on page 6 towards the end of the HFNC section.

C5: It would be helpful, if the authors could specify their sentence on page 4: “It is beneficial to administer surfactant early (before 2 hours of life) when infant requires escalation of the support.” and add a reference.

R5: Thanks for the input. We have now modified this sentence to provide more clarity and added reference. (Page 4 under initiation of NIPPV).

C6: Page 2: The sentence “Routine intubation of extremely preterm infants has given the way for early use of nasal continuous positive airway pressure (NCPAP) starting in the delivery room and selective surfactant use, after large trials showed a benefit with such an approach [11,12].” does not come clear-please rephrase.

R6: We have now modified the sentence. (Page 2).

C7: The authors should avoid to name specific respirators and highflow devices but should instead specify the mode of action of the different devices to stay product independent.

R7: Thanks for the suggestion. We have now deleted the device names and emphasized the mechanisms of action. (End of Page 2 and beginning of Page 3).

Reviewer 2 Report

Excellent work.  I enjoyed reading this review. 

My only suggestion is to add a brief explanation of the importance of bedside skills.  For example, cited trials on CPAP showed an incidence of BPD of 31%-33% and failure rate of 50%-80%.  However, there are experienced centers that reported BPD incidence of 5%-6% (using the same definition), and a failure rate <10%.  Therefore, there should be an emphasis on providing proper training and developing excellent caregiver skills that will decrease failure rate and prevent BPD.  I will be happy to discuss offline. 

Otherwise the review is an excellent resource. 

Best regards,

Hany Aly ([email protected])   

Author Response

Excellent work.  I enjoyed reading this review. 

C1: My only suggestion is to add a brief explanation of the importance of bedside skills.  For example, cited trials on CPAP showed an incidence of BPD of 31%-33% and failure rate of 50%-80%.  However, there are experienced centers that reported BPD incidence of 5%-6% (using the same definition), and a failure rate <10%.  Therefore, there should be an emphasis on providing proper training and developing excellent caregiver skills that will decrease failure rate and prevent BPD.  I will be happy to discuss offline. 

Otherwise the review is an excellent resource. –

R1: The authors would like to thank the reviewer for offering this important suggestion. We have now included this part in the manuscript. (Page 3).

Reviewer 3 Report

The paper by Dumpa and Bhandari is a comprehensive review about possible non-invasive ventilatory modalities to prevent BPD development.

This is a well written review, some comments to even improve it.

Comments

  • CPAP section: authors should analyze deeper pressure levels and possible evidence regarding BPD. There is still an open debate about “high” and “low” pressures In the acute setting, but also the long term outcome should be evaluated.
    • The cited reference n° 12 (COIN Trial) about higher levels of PEEP being related to higher incidence of PNX (5th line from the bottom of page 4) says: “Our finding that infants in the CPAP group had more pneumothoraxes reflects the results of other studies. Of the infants with a pneumothorax, 96% underwent ventilation. The airway pressure when the pneumothorax was diagnosed was not recorded. The median CPAP pressure on admission was 8 cm of water for infants in whom a pneumothorax developed as well as for those in whom a pneumothorax did not develop.”. I suggest to find a better reference to justify that statement.
  • NIPPV section: authors should better explore the possible factors explaining the superiority of SNIPPV over NIPPV, see:
    • Moretti C, Gizzi C, Montecchia F, Barbàra CS, Midulla F, Sanchez-Luna M, Papoff P. Synchronized Nasal Intermittent Positive Pressure Ventilation of the Newborn: Technical Issues and Clinical Results. 2016;109(4):359-65. doi: 10.1159/000444898. Epub 2016 Jun 3. PMID: 27251453.
    • Moreau-Bussière F, Samson N, St-Hilaire M, Reix P, Lafond JR, Nsegbe E, Praud JP. Laryngeal response to nasal ventilation in nonsedated newborn lambs. J Appl Physiol (1985). 2007 Jun;102(6):2149-57. doi: 10.1152/japplphysiol.00891.2006. Epub 2007 Mar 1. PMID: 17332270.
    • Hadj-Ahmed MA, Samson N, Bussières M, Beck J, Praud JP. Absence of inspiratory laryngeal constrictor muscle activity during nasal neurally adjusted ventilatory assist in newborn lambs. J Appl Physiol (1985). 2012 Jul;113(1):63-70. doi: 10.1152/japplphysiol.01496.2011. Epub 2012 Apr 19. PMID: 22518828.
  • View that NCPAP systems have been described, another interesting aspect to completely explore non-invasive ventilation is the delivering interface.

In conclusion this review deserves publication, with minimal possible improvements.

Author Response

Comments and Suggestions for Authors

The paper by Dumpa and Bhandari is a comprehensive review about possible non-invasive ventilatory modalities to prevent BPD development.

This is a well written review, some comments to even improve it.

Comments

  • C1: CPAP section: authors should analyze deeper pressure levels and possible evidence regarding BPD. There is still an open debate about “high” and “low” pressures In the acute setting, but also the long term outcome should be evaluated.

R1: Thank you for the suggestion. We have now added two studies evaluating different levels of PEEP in preterm infants with RDS. (Page 2).

  • C2: The cited reference n° 12 (COIN Trial) about higher levels of PEEP being related to higher incidence of PNX (5th line from the bottom of page 4) says:Our finding that infants in the CPAP group had more pneumothoraxes reflects the results of other studies. Of the infants with a pneumothorax, 96% underwent ventilation. The airway pressure when the pneumothorax was diagnosed was not recorded. The median CPAP pressure on admission was 8 cm of water for infants in whom a pneumothorax developed as well as for those in whom a pneumothorax did not develop.”  I suggest to find a better reference to justify that statement.

R2: Thanks for bringing this to our attention. We have now included another study that have shown pneumothoraces associated with use of higher levels of NCPAP. (Page 5).

  • C3: NIPPV section: authors should better explore the possible factors explaining the superiority of SNIPPV over NIPPV, see:
    • Moretti C, Gizzi C, Montecchia F, Barbàra CS, Midulla F, Sanchez-Luna M, Papoff P. Synchronized Nasal Intermittent Positive Pressure Ventilation of the Newborn: Technical Issues and Clinical Results. 2016;109(4):359-65. doi: 10.1159/000444898. Epub 2016 Jun 3. PMID: 27251453.
    • Moreau-Bussière F, Samson N, St-Hilaire M, Reix P, Lafond JR, Nsegbe E, Praud JP. Laryngeal response to nasal ventilation in nonsedated newborn lambs. J Appl Physiol (1985). 2007 Jun;102(6):2149-57. doi: 10.1152/japplphysiol.00891.2006. Epub 2007 Mar 1. PMID: 17332270.
    • Hadj-Ahmed MA, Samson N, Bussières M, Beck J, Praud JP. Absence of inspiratory laryngeal constrictor muscle activity during nasal neurally adjusted ventilatory assist in newborn lambs. J Appl Physiol (1985). 2012 Jul;113(1):63-70. doi: 10.1152/japplphysiol.01496.2011. Epub 2012 Apr 19. PMID: 22518828.
  • View that NCPAP systems have been described, another interesting aspect to completely explore non-invasive ventilation is the delivering interface.

R3: Thank you for the input. We have now included more details on synchronization of NIPPV and nasal interfaces. (Pages 3,4).

C4: In conclusion this review deserves publication, with minimal possible improvements.

R4: Thank you for the positive comment.